# Primary Stability of Revision Acetabular Reconstructions Using an Innovative Bone Graft Substitute: A Comparative Biomechanical Study on Cadaveric Pelvises

**DOI:** 10.3390/ma13194312

**Published:** 2020-09-27

**Authors:** Federico Morosato, Francesco Traina, Ronja A. Schierjott, Georg Hettich, Thomas M. Grupp, Luca Cristofolini

**Affiliations:** 1Department of Industrial Engineering, Alma Mater Studiorum, Università di Bologna, 40131 Bologna, Italy; federico.morosato2@unibo.it; 2Chirurgia Protesica, IRCCS Rizzoli Orthopaedic Institute, 40136 Bologna, Italy; traina.francesco@gmail.com; 3Department of Biomedical and Neuromotor Sciences, Università di Bologna, 40126 Bologna, Italy; 4Aesculap AG, Research & Development, Am Aesculap-Platz, 78532 Tuttlingen, Germany; ronja_alissa.schierjott@aesculap.de (R.A.S.); georg.hettich@aesculap.de (G.H.); thomas.grupp@aesculap.de (T.M.G.); 5Department of Orthopaedic Surgery, Physical Medicine and Rehabilitation, Campus Grosshadern, Ludwig Maximilian University, 81377 Munich, Germany

**Keywords:** revision hip surgery, severe contained acetabular defects, synthetic bone graft substitute, in vitro stability, biomechanical testing

## Abstract

Hip implant failure is mainly due to aseptic loosening of the cotyle and is typically accompanied by defects in the acetabular region. Revision surgery aims to repair such defects before implantation by means of reconstruction materials, whose morselized bone graft represents the gold standard. Due to the limited availability of bone tissue, synthetic substitutes are also used. The aim of this study was to evaluate if a synthetic fully resorbable tri-calcium phosphate-based substitute can provide adequate mechanical stability when employed to restore severe, contained defects, in comparison with morselized bone graft. Five cadaveric pelvises were adopted, one side was reconstructed with morselized bone graft and the other with the synthetic substitute, consisting of dense calcium phosphate granules within a collagen matrix. During the biomechanical test, cyclic load packages of increasing magnitude were applied to each specimen until failure. Bone/implant motions were measured through Digital Image Correlation and were expressed in terms of permanent and inducible translations and rotations. The reconstruction types exhibited a similar behavior, consisting of an initial settling trend followed by failure as bone fracture (i.e., no failure of the reconstruction material). When 2.2 Body Weight was applied, the permanent translations were not significantly different between the two reconstructions (*p* = 0.06–1.0) and were below 1.0 mm. Similarly, the inducible translations did not differ significantly (*p* = 0.06–1.0) and were below 0.160 mm. Rotations presented the same order of magnitude but were qualitatively different. Overall, the synthetic substitute provided adequate mechanical stability in comparison with morselized bone graft, thus representing a reliable alternative to treat severe, contained acetabular defects.

## 1. Introduction

About 10–15% of hip replacements performed worldwide every year are related to revision surgery [1,2,3,4,5]. Aseptic loosening of the cotyle is the main cause of hip implant failure and is generally accompanied with bone loss and generation of defects in the acetabular region [6], which must be addressed prior implantation to achieve initial and long-term stability. Defect restoration is typically addressed by adding a reconstruction material in the impaired region. In the case of contained defects, the gold standard is represented by impacted morselized bone graft, but, due to its limited availability, synthetic materials are also adopted as alternative solutions [7,8,9,10]. Among these, calcium phosphate CaP-based materials such as hydroxyapatite (HA) or tri-calcium-phosphate (TCP) represent reliable options, due to high availability, biocompatibility and osteoconductive properties [11].

The achievement of adequate initial stability is crucial to granting long-term stability of the revision reconstruction. For this reason, primary stability typically represents the principal focus of biomechanical studies investigating the mechanical response of acetabular reconstructions when subjected to post-operative loading (which is cyclic by nature).

Arts et al. (2006) investigated the impact of the size and of the washing of morselized bone graft, used to restore segmental defects (similar to AAOS Type III), on the stability of acetabular reconstructions [12]. Bolder et al. (2003) assessed experimentally, in simple acetabular models, the primary stability of revision reconstructions addressed with bone graft and TCP/HA granules, aiming to identify the combination that allowed optimal cement penetration [13]. A similar approach was adopted by other authors that investigated the mechanical stability of acetabular reconstructions obtained via impaction grafting technique and cemented cups [14,15,16]. In all the studies above, biomechanical analysis was performed on synthetic models of acetabular defects, treated with resorbable materials in combination with bone cement.

Recently, a bone graft substitute made of TCP tetrapods has been shown to grant adequate mechanical load capacity in a simplified bone defect model [17]. In a previous study, the use of a bone graft substitute made of molded bodies within a matrix material was proven to grant implant stability, when used to restore bone defects with a standardized geometry implemented in a foam model [18]. In both the above-mentioned studies, synthetic acetabular models were used.

To the best of the Author’s knowledge in only one in vitro study by Jacofsky et al. (2012) morselized bone graft was compared with a resorbable synthetic CaP-based material adopted to reconstruct a cavitary defect implemented in a cadaveric model [19].

The aim of the present study was to evaluate the potentiality of a fully resorbable CaP-based bone graft substitute for the treatment of severe, contained acetabular defects. The synthetic material proposed in Hettich et al. (2019) [17] was modified by embedding the TCP tetrapods in a collagen matrix so as to enhance its handling properties. The bone substitute resulted in a moldable CaP-putty. We hypothesized that the CaP-putty would grant a primary stability comparable to that which conventional impaction technique with bone graft provides. Measurements of primary stability, expressed in terms of implant permanent migration and inducible micromotions, were used to compare the two reconstruction techniques. Implant motions were measured by mean of Digital Image Correlation (DIC).

## 2. Materials and Methods

### 2.1. Ethic Statement

Five pairs of fresh-frozen hemipelvises were obtained through ethically approved donation programs (Table 1). This study was authorized by the Bioethics Committee of the University of Bologna (Prot. 179,610 of 7 December 2018). No information about donor’s laterality was available.

### 2.2. Preparation of the Specimens

The bones were wrapped in cloths soaked with physiological saline solution when not in use and thawed at room temperature prior to testing. The soft tissues in the periacetabular region were removed so as to expose the anatomical landmarks necessary for the alignment procedure. Each hemipelvis was aligned in a reliable reference frame and potted in correspondence to the sacro-iliac joint in an aluminum pot with bone cement [20].

To replicate consistent defects in each hemipelvis, a standardized protocol for defect implementation, based on statistical shape modeling and quantitative defect analysis, was adopted [18,21]. Critical, contained defects were replicated representing mostly medial defects with rim damage of approximately one third of the circumference, located at the inferior-posterior area of the rim (Figure 1). To enhance consistency, each defect was scaled on anatomical features of the native acetabulum: the thickness of the acetabular roof, the thickness of the anterior column and the thickness of the posterior wall.

Each hemipelvis and its contralateral side were randomly selected to be treated with CaP-putty or with morselized bone graft (Figure 1). The acetabular reconstructions were performed by an experienced surgeon. Following the indications of the surgeon, the morselized bone graft were prepared from a proximal femoral epiphysis using a rongeur, in order to achieve a medium diameter of abou0t 8mm. The bone graft was impacted into the defect cavity by gently hammering with a dedicated hemispherical impactor and added until the contained defect was fulfilled and level with the surface of the acetabulum. The same procedure was used with the CaP-putty.

The CaP-putty consisted of TCP tetrapods within a collagen matrix. The putty should provide a loadable, osteoconductive, and osteoactive putty to reconstruct large acetabular bone defects. A recent study showed the load capacity and the osteoconductive properties of the tetrapods without collagen [17]. To improve the handling properties and to include a kind of osteoactive stimulus, the tetrapods were embedded within a collagen matrix in a weight ratio of 95% tetrapods and 5% collagen. A tetrapods/collagen slurry was prepared and filled in cavities with a length of 5 cm. Using lyophilization, bars of tetrapod/collagen bars were created. In contact with water, the bars become moldable and were used in the present study to fill the defect in form of a CaP-putty. The collagen matrix collapses when the putty is impacted and the tetrapods are in contact with each other. After resorption of the collagen, an osteoconductive scaffold for bone ingrowth is expected to remain.

Commercial cups (Plasmafit PLUS 7, Aesculap, Tuttlingen, Germany) were press fitted into the acetabular cavity and two anchoring screws were inserted medially to enhance stability (Figure 1). Whenever possible, the same cup size was implanted in the contralateral hemipelvises (in three cases a discrepancy of one size was required to accommodate for the asymmetry of the same donor).

To allow the DIC software to correlate, a high-contrast black-on-white speckle pattern was applied on the specimen surface.

### 2.3. Biomechanical Testing

A validated protocol was used for the biomechanical testing [22]. Walking was selected among the recommended activities for post-op rehabilitation. In particular, the direction of the peak force measured in vivo during level walking was extracted from open datasets [23] and a dedicated mechanical setup was produced so as to apply the force in the selected direction during the biomechanical test. A servo-hydraulic testing machine was used to load the specimens. The test consisted of 50-cycles load packages with increasing amplitude. The first load package (up to 1 BW) served as pre-conditioning. Then, the biomechanical test was extended, increasing the load until visible specimen failure (Figure 2).

### 2.4. Measurement of Implant Motion

A commercial DIC system (Q400, Dantec Dynamics, Skovlunde, Denmark) was used to measure the motions of implant and bone. Two cameras (5 MegaPixels, 2440 × 2050 pixels, 8-bit) equipped with high-quality metrology-standard 17 mm lenses (Xenoplan, Schneider-Kreuznach, Bad Kreuznach, Germany) were used to obtain 3D measurements. The cameras were positioned so as to frame the implant and the superior aspect of the acetabulum (Figure 2).

To compute the bone/implant motions (translations and rotations along/about cranio-caudal, antero-posterior and medio-lateral axes), the DIC-measured displacements were post-processed through a dedicated script in Matlab (2017 Edition, MathWorks, Natick, MA, USA) [22]. In particular, the permanent migration (i.e., the migration accumulated cycle after cycle), and the inducible micromotion (i.e., the recoverable motion between load peak and valley) were analyzed.

### 2.5. Statistical Analysis

To exclude outliers, the Peirce’s criterion was applied. To assess if the effects on implant motions deriving from the two bone reconstruction materials were statistically different, a Wilcoxon signed rank test was performed at each load level. All statistical tests were performed with Matlab.

## 3. Results

The specimens failed at different load magnitudes, between 2.2 and 5.0 BW, due to fracture of the posterior column To compare the effect of the bone reconstruction material on implant stability, the results will be presented up to 3 BW, which corresponds to the magnitude of the peak load measured in vivo in the hip during walk [23]. This force magnitude was reached before bone fracture in all specimens but one: in this case all the load packages preceding fracture were used for the comparison with the other specimens.

The permanent migration steadily increased throughout the test as the load increased (Figure 3). Migration showed a visible settling trend from the beginning to the end of each single load package (of 50 cycles each) at low forces, whereas migrations kept growing during each of the last load packages (higher forces). At 2.2 BW (all specimens reached this level), the resultant permanent migration for the specimens with bone graft was 0.57 mm (median; range: 0.12–2.26 mm) and with CaP-putty was 0.95 mm (median; range: 0.33–2.67 mm). At 3.0 BW the permanent migration (for the four specimens that reached 3 BW) with bone graft was 0.84 mm (median; range: 0.26–2.67 mm) and with CaP-putty was 1.39 mm (median; range: 0.69–3.43 mm). The difference of migrations between the two groups was not statistically significant for any load level (Wilcoxon signed rank sum test, *p* = 0.06–1.0 for the different load packages).

The inducible micromotions had a more irregular trend (Figure 3). They were generally constant or slightly decreasing within the same load package for lower loads. When the implant started migrating, the micromotions also fluctuated within the same load package. Large inducible micromotions were typically followed by larger permanent migrations. At 2.2 BW the inducible micromotions for the specimens with bone graft were 0.11 mm (median; range: 0.06–0.21 mm) and with CaP-putty were 0.16 mm (median; range: 0.05–0.32 mm). At 3.0 BW the micromotions (for the four specimens that reached 3 BW) with bone graft were 0.15 mm (median; range: 0.15–0.43 mm) and with CaP-putty were 0.24 mm (median; range: 0.05–0.53 mm). The difference of inducible micromotions between the two groups was not statistically significant for any load level (Wilcoxon signed rank sum test, *p* = 0.06–1.0 for the different load packages).

Looking at the single motion direction, the largest permanent translation occurred along the medio-lateral axis for both the reconstruction techniques (Figure 4 and Figure 5). At 3.0 BW, the permanent translations along the cranio-caudal and the antero-posterior axes were on average respectively 31% and 47% of the translation along the medio-lateral axis for reconstructions with bone graft and 24% and 87% for reconstructions with CaP-putty. The largest component of permanent rotation was about the antero-posterior axis (i.e., change of inclination).

The inducible translations and rotations had differences between components similar to those observed for the permanent ones, but varied significantly between specimens and during the test (Figure 4 and Figure 5).

## 4. Discussion

The aim of the present study was to evaluate the potentiality of a fully resorbable CaP-based bone graft substitute in granting adequate mechanical stability to acetabular reconstruction, in comparison with impaction bone grafting technique, assumed as gold standard.

Digital image correlation was used to measure the bone/implant motions during the application of cyclic load of increasing magnitude. In all tests, specimen failure consisted of the fracture of the posterior column, reached at different load magnitudes, and was never due to excessive deformation or failure of the reconstruction material.

At 2.2 BW the resultant permanent migration never exceeded 1.0 mm, thus were below the clinical threshold associated with late implant loosening [24]. At the same load magnitude, inducible micromotions never exceeded 160 micrometers, close to the value generally assumed to grant osteointegration in real bone [25]. It must be noticed that the inducible migration measured is not just interface micromotion, but is partly due to the deformation of the graft material. Therefore, the actual interface micromotions were definitely lower. Most of the permanent migration occurred within the first cycles of each load package up to 2.2 BW, thus exhibiting a settling trend within each load package. Such tendency to settle was confirmed by the trend of the inducible micromotions, which were generally constant or slightly decreasing within the same load package. Specimens reconstructed with the CaP-putty exhibited larger permanent migrations and inducible micromotions, in comparison to specimens reconstructed with morselized bone graft. However, this difference was not statistically significant (*p* = 0.06–1.0).

In past studies, investigating the mechanical stability of reconstructed acetabula, the average measured cup migration had the same order of magnitude as the one in the present study, but larger cup motions were associated with defects reconstructed with morselized bone graft [12,13,15,16]. Such difference may be related to the defect model, the reconstruction techniques and the test methods used to assess the acetabular stability. In most studies central, cavitary, contained defects were implemented (similar to AAOS Type 2 defects), while in our study the defect also involved the posterior column significantly. For this reason, even if the largest migration was measured along the medio-lateral axis (i.e., in line with all of the studies mentioned above), a large cup migration was also measured in the direction of the posterior column. In most studies cups were cemented within a layer of reconstruction material (whether bone graft or synthetic), used to fill the cavity, while in the present study, cups were press fitted, and two screws were added to enhance the stability. A full comparison may be done only with Jacofsky et al. (2012). In that study, a loading direction comparable with the one adopted in the present study (i.e., simulating level walking) was applied to cadaveric hemipelvises reconstructed with morselized bone graft or with a bioresorbable calcium phosphate injectable material. The accumulated cup migration was similar between the two reconstruction types, i.e., it was larger with the bone graft but the difference with the synthetic material was less than 20 micrometers at the end of the test. In addition, the settling trend was in line with our study, consisting of a progressive settlement of the bone graft during the test; however, the synthetic material used in that study exhibited an increasing rate of migration [19].

It is recognized that the preparation, size, and size distribution of bone grafts can affect their behavior. To address this, the bone chips were prepared according to the instructions of a senior hip revision surgeon (FT) to fit the characteristics of clinically applied bone grafts. With a mean size of 8 mm, most of the bone chips were larger than the β-TCP granules with an edge length of 3.3 mm. This may have affected the primary stability results, especially in the context of the very thin medial wall in the present study, i.e., the fact that only the periosteum was left in some cases: The smaller, relatively pointed granules may have pushed more against the periosteum than the larger, smoother grafts which may have rather interlocked to a meshwork covering the periosteum and hence reducing relative motion. However, the granule-size was chosen based on pre-tests and in order to provide satisfying handling properties, including the possibility to fill smaller defects, while still providing large enough inter-granule pores (i.e., the space between the granules) and sufficient pore interconnectivity to enable bone ingrowth [17].

Some limitations of the present work must be mentioned. The sample size (5 pairs) was rather small. This was justified by previous investigations. In particular, our study relied on an extensively validated testing method based on Digital Image Correlation, which provided very small measurement errors [22]. To overcome the inter-specimen variability, paired hemipelvises were used, allowing more powerful statistical analysis based on paired data. For these reasons, five pairs of hemipelvises provided enough statistical power. The sample size was limited due to specimen availability. However, as the test was devised to compare two reconstruction techniques, paired specimens were adopted; thus, providing a reliable comparison. Moreover, the number of the specimen is comparable with the sample size of other biomechanical analyses related to implant stability [19,26,27]. The in vitro behavior of cadaveric specimens is different from living tissue. However, as the specimens were fresh-frozen, and hydration was granted during preparation, preservation and test, the mechanical properties were not compromised [28]. As in the cadaver experimentation tissue response (e.g., adaptation, ingrowth, inflammation) is not present, we could only investigate a post-operative condition. The same applies to the expected graft resorption over time: due to the putty preparation with water and the short exposure time to a very limited amount of body fluid, it is not expected that the moisture has a considerable influence on the putty characteristics or performance in the timeframe of the in vitro test, which are therefore representative of the post-operative condition.

The loading configuration was limited to a single direction, reducing the complexity of forces and moments acting in the acetabulum. The loading configuration was limited to a single direction, reducing the complexity of forces and moments acting in the acetabulum. Such an approach, already adopted by most authors, has been proved to induce cup motions comparable with ones clinically observed while granting a good robustness and reliability to the results (i.e., minimizing the sources of experimental error [22]). Moreover, as paired tests were performed, the effect of the loading configuration on the acetabular stability was reduced in comparison with the effect of the reconstruction material. The current study focused on contained defects, thus, in case of different defect models (e.g., segmental or uncontained), results may be different and further investigations may be required.

Overall, the study demonstrated that the resorbable CaP-putty bone substitute proposed in the present study granted a mechanical stability comparable with the gold standard technique (impaction bone grafting) when applied in the acetabulum with severe, contained defects. The material was tested until failure of the specimen, thus over 3 BW in most cases, thus representing a critical peak load for a patient undergoing revision surgery. For lower forces (more reasonable for revision patients) the difference from the morselized bone graft was minimal. The clinical thresholds related to implant failure (i.e., 1 mm for the permanent migration and 150 micrometers for the inducible motions) were not exceeded. Such results confirmed the good load capacity of the proposed material [17], overcoming the lack of mechanical stability that calcium-based materials generally have. In conclusion, this study has shown the newly developed CaP-putty has very promising biomechanical properties, when applied to severe contained defects, with remarkable advantages in terms of cost, availability, conservation, and body-rejection.

## Figures and Tables

**Figure 1 materials-13-04312-f001:**
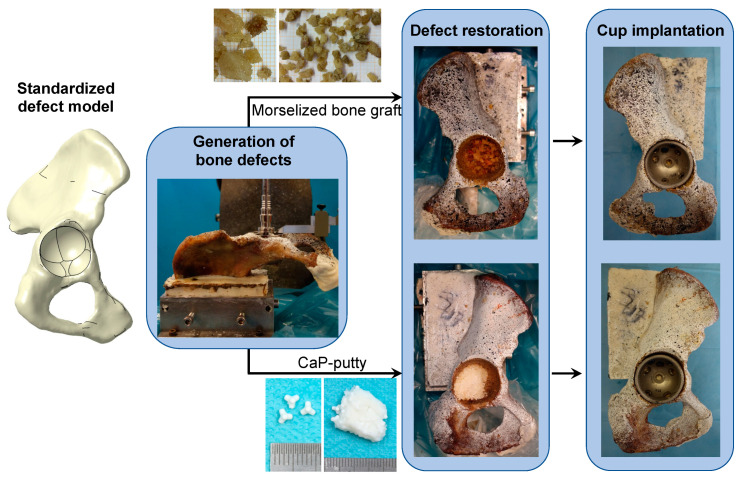
Preparation of the specimens: The standardized acetabular defect was implemented based on the statistical shape model developed in Schierjott et al. [21]. The hemipelvis was machined in consecutive steps using different commercial surgical reamers, with reproducible defects scaled according to the specimen-specific dimensions. Each hemipelvis and its contralateral side (*n* = 10 hemipelvises in total) were alternatively reconstructed with morselized bone graft or CaP-based material. Commercial cups were press-fit in the acetabular cavity and two anchoring screws were used.

**Figure 2 materials-13-04312-f002:**
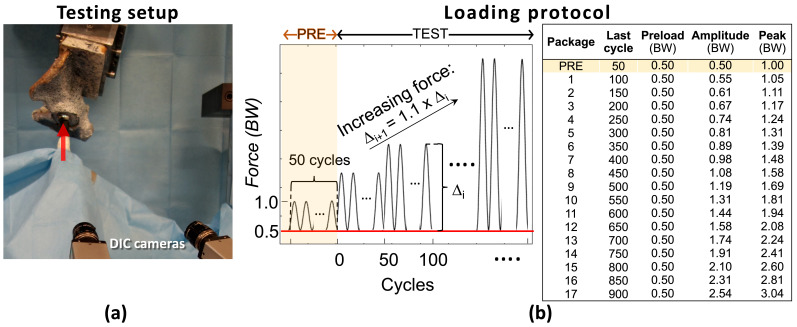
Test setup and loading protocol: (**a**) The specimen was aligned in the testing frame so as to apply a force (arrow) in a direction that replicated level walking. The cameras of the DIC system were placed so as to frame both the cup and the surrounding bone. (**b**) Load cycles of increasing magnitude were applied: each load packages consisted of 50 cycles; each package was 10% larger than the previous one; the force was scaled on the patient body weight (BW). The first load package (50 cycles 1.00 BW) was used for pre-conditioning before the actual test.

**Figure 3 materials-13-04312-f003:**
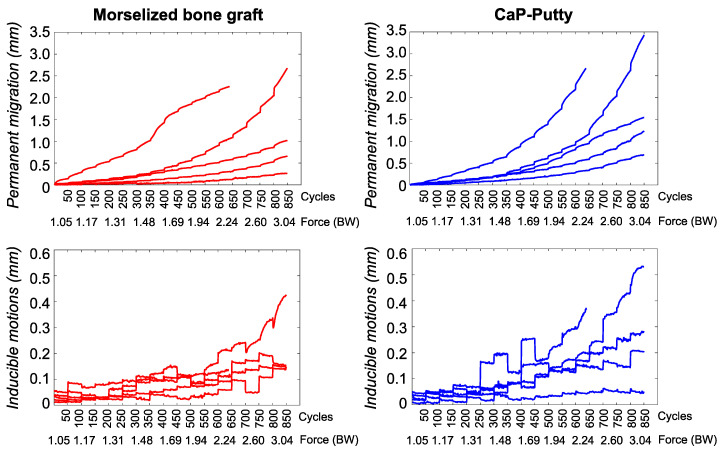
Morselized bone graft vs. CaP-putty: The resultant permanent migration and the resultant inducible micromotions measured during each test are shown for the morselized bone graft (**left**, *n* = 5 specimens) and CaP-putty (**right**, *n* = 5 contralateral specimens). On the *x*-axis the load cycles (in packages of 50 cycles) and the corresponding magnitude of the applied force are indicated. Motions during the preconditioning load package are not presented. Not-significant differences were found between the two groups (Wilcoxon signed rank sum test, *p* = 0.06–1.0 for the different load packages).

**Figure 4 materials-13-04312-f004:**
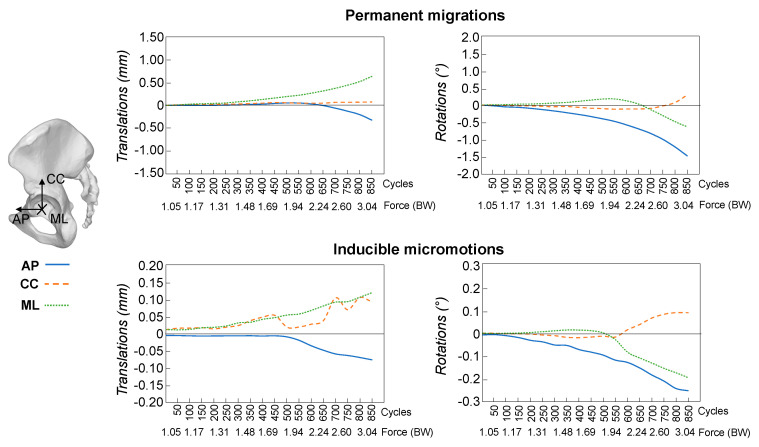
Morselized bone graft: The permanent migrations and inducible micromotions throughout the test are presented as the median trend of the specimens (*n* = 5 hemipelvises). For the translations, the components along the antero-posterior (AP), cranio-caudal (CC) and medio-lateral (ML) directions are reported. The rotations are reported in terms of individual components about the antero-posterior (AP), cranio-caudal (CC) and medio-lateral (ML) axis. On the *x*-axis the load cycles (in packages of 50 cycles) and the corresponding magnitude of the applied force are indicated. Motions during the preconditioning load package are not presented.

**Figure 5 materials-13-04312-f005:**
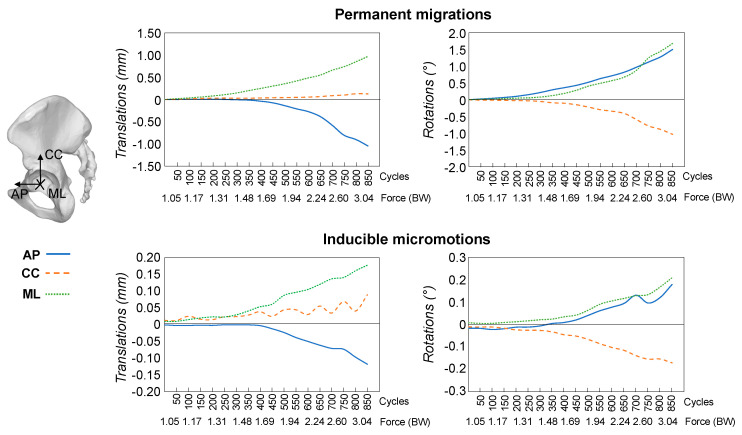
CaP-putty: The permanent migrations and inducible micromotions throughout the test are presented as the median trend of the specimens (*n* = 5 hemipelvises). For the translations, the components along the antero-posterior (AP), cranio-caudal (CC) and medio-lateral (ML) directions are reported. The rotations are reported in terms of individual components about the antero-posterior (AP), cranio-caudal (CC) and medio-lateral (ML) axis. On the *x*-axis the load cycles (in packages of 50 cycles) and the corresponding magnitude of the applied force are indicated. Motions during the preconditioning load package are not presented.

**Table 1 materials-13-04312-t001:** List of specimens, including the donors’ details, and the size of the implanted cups. The last column reports the difference between the two reconstruction materials for the acetabular defects.

Donor	Cause of Death	Sex	Age (Years)	Height (cm)	Body Weight (kg)	BMI (kg/m^2^)	Side	Primary Cup Size (mm)	Revision Cup Size (mm)	Reconstruction Material
**#1**	Sepsis	F	83	164	63	23	L	56	58	Bone graft
							R	56	58	CaP-putty
**#2**	Respiratory paralysis	M	70	175	79	26	L	52	54	Bone graft
							R	54	56	CaP-putty
**#3**	Unknown	M	74	176	78	25	L	48	50	CaP-putty
							R	48	50	Bone graft
**#4**	Coronary thrombosis	M	71	187	92	26	L	60	62	Bone graft
							R	62	64	CaP-putty
**#5**	Cardiac arrhythmia	M	61	181	96	29	L	56	58	Bone graft
							R	54	56	CaP-putty
Median		-	71	176	79	26	-	55	56.6	5 vs. 5
SD		-	7.9	8.5	13.2	2.2	-	4.5	4.3	-

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
