# Peer review of "Primary Stability of Revision Acetabular Reconstructions Using an Innovative Bone Graft Substitute: A Comparative Biomechanical Study on Cadaveric Pelvises"

_materials, 2020, doi:10.3390/ma13194312_

Round 1

Reviewer 1 Report

In this research work, Morosato et al. performed revision acetabular reconstructions using an innovative bone graft substitute and tested its primary stability to identify a synthetic alternative to morselized bone grafts to reconstruct severe contained acetabular defects. The study demonstrated that the resorbable CaP putty bone graft substitute provides a mechanical stability comparable to that of the gold standard technique morselized bone graft in an acetabulum with severe, contained defects.

However, the data and the details of the figure plates and the legends are not clearly presented. The figure legends for Figure 1-4 should be presented with a title. The number of replicates is skipped in some of the figure legends. Each figure and the corresponding legends should also describe the data, mathematical and statistical notations. 

Since the authors have tested the implant only in a model for contained defects, the results can not be generalized for acetabular defects in general. The authors need to include future studies with other models of acetabular defects.

Author Response

See attached PDF

Reviewer 2 Report

The manuscript "Primary stability of revision acetabular reconstructions using an innovative bone graft substitute: A comparative biomechanical study on cadaveric pelvises" is interesting and with new data, however before acceptance this manuscript needs revision.

The main weak point of this manuscript is the limitation of the number of the samples.

Minor attentions:

page 2 line 54: the reference Ars et at was cited, please add a year of the publication. 

page 2 line 55: the reference Bolder et al. - please add a year of the publication

page 2 line 82 : "This Study..." change as: "This study..."

Reference list should be checked carefuly and the style of the references should be unified.

Author Response

See attached PDF

Reviewer 3 Report

When authors used "significant", it should accompany with p-value. However, it was missing in the abstract.

In methods, "The level of significance was p=0.05 for all analysis" was incorrect. 

Current model is a study in cadaver. After death, bone is brittle and there is no biological response such as inflammation. Thus, it should be discussed in the limitations of this study. 

Please check ref. 1 to 4. It does not have detailed information. If it is possible, please replace them to the published articles.

Author Response

See attached PDF

Reviewer 4 Report

The authors conducted research on the primary mechanical stability of calcium-phosphate putty bone substitute for reconstructing the acetabular bone defects. While the research path is straight forward and well-organized, I have some minor concerns that require modifications.  

1. Methods

a. Do you think that the size of particulates either in bone graft or bone substitute groups can influence your findings? How about the effect of moisture on CaP putty? how did you address these factors?

2. Discussion

What do you think about the impact of cohesive strength or the inter particulate adhesion of CaP putty on your findings? 

4. References  

Please modify the reference style according to the journal's guidelines (e.g. Ref #6, 7, 9, 22, 25). 

Author Response

See attached PDF
